# SNPs in Genes Related to DNA Damage Repair in Mycobacterium Tuberculosis: Their Association with Type 2 Diabetes Mellitus and Drug Resistance

**DOI:** 10.3390/genes13040609

**Published:** 2022-03-29

**Authors:** Damián E. Pérez-Martínez, Gustavo A. Bermúdez-Hernández, Carlos F. Madrazo-Moya, Irving Cancino-Muñoz, Hilda Montero, Cuauhtemoc Licona-Cassani, Raquel Muñiz-Salazar, Iñaki Comas, Roberto Zenteno-Cuevas

**Affiliations:** 1Programa de Doctorado en Ciencias de la Salud, Instituto de Ciencias de la Salud, Universidad Veracruzana, Av. Luis, Dr. Castelazo Ayala s/n, Col. Industrial Animas, Xalapa 91190, Mexico; biol_dam_perez@hotmail.com (D.E.P.-M.); dcsgabh@gmail.com (G.A.B.-H.); 2Biomedical Institute of Valencia IBV-CSIC, C. de Jaume Roig, 11, 46010 Valencia, Spain; cf.madrazo@gmail.com (C.F.M.-M.); icancino@ibv.csic.es (I.C.-M.); icomas@ibv.csic.es (I.C.); 3CIBER of Epidemiology and Public Health, 08908 Madrid, Spain; 4Instituto de Salud Pública, Universidad Veracruzana, Av. Luis Castelazo Ayala s/n, A.P. 57, Col. Industrial Animas, Xalapa 91190, Mexico; hildaisp@gmail.com; 5Escuela de Ingeniería y Ciencias, Tecnológico de Monterrey, Ave. Eugenio Garza Sada 2501 Sur, Monterrey 64849, Mexico; clicona@tec.mx; 6Red Multidisciplinaria de Investigación en Tuberculosis, Mexico City 14080, Mexico; ramusal@uabc.edu.mx; 7Division of Integrative Biology, The Institute for Obesity Research, Tecnológico de Monterrey, Monterrey 64849, Mexico; 8Laboratorio de Epidemiología y Ecología Molecular, Escuela de Ciencias de la Salud, Universidad Autónoma de Baja California, Ensenada 22890, Mexico

**Keywords:** tuberculosis, diabetes mellitus, resistance, DNA repair

## Abstract

Genes related to DNA damage repair in Mycobacterium tuberculosis are critical for survival and genomic diversification. The aim of this study is to compare the presence of SNPs in genes related to DNA damage repair in sensitive and drug-resistant M. tuberculosis genomes isolated from patients with and without type 2 diabetes mellitus (T2DM). We collected 399 M. tuberculosis L4 genomes from several public repositories; 224 genomes belonging to hosts without T2DM, of which 123 (54.9%) had drug sensitive tuberculosis (TB) and 101 (45.1%) had drug resistance (DR)-TB; and 175 genomes from individuals with T2DM, of which 100 (57.1%) had drug sensitive TB and 75 (42.9%) had DR-TB. The presence of SNPs in the coding regions of 65 genes related to DNA damage repair was analyzed and compared with the resistance profile and the presence/absence of T2DM in the host. The results show the phylogenetic relationships of some SNPS and L4 sub-lineages, as well as differences in the distribution of SNPs present in DNA damage repair-related genes related to the resistance profile of the infecting strain and the presence of T2DM in the host. Given these differences, it was possible to generate two discriminant functions to distinguish between drug sensitive and drug resistant genomes, as well as patients with or without T2DM.

## 1. Introduction

With more than 10 million cases and 1.2 million deaths, tuberculosis (TB) remains an infectious disease of global importance [1]. Currently, drug resistance (DR), HIV, and type 2 diabetes mellitus (T2DM) are elements that contribute to the status of TB as a severe health problem. In this regard, T2DM leads to a 3.1 (2.27–4.26) fold increased risk of developing TB [2], through reactivation of latent infection or progression of recent infection [3], and the TB-T2DM binomial has been identified as a risk factor for unfavorable treatment outcomes, increasing the probability of failure, relapse, or death [4]. Patients with TB-T2DM are also more frequently found within the recent TB transmission groups [5], having a 4.7 (1.4–11.3) fold increased risk of becoming single-drug resistant, and a 2.8 (2.2–3.4) to 3.5 (1.1–11.1) fold increased risk of becoming multi-drug resistant (TB-MDR) [6,7]. This behavior has been explained as a consequence of the decrease in plasma concentrations of anti-tuberculosis drugs [8] and the interference from medications taken for glycemic control [9]. Besides, high glucose concentrations and immunological alterations in patients with T2DM [10,11,12] have been shown to promote the elongation of active infection and increased proliferation of TB bacilli [13].

From a molecular perspective, Single Nucleotide Polymorphisms (SNPs) are the primary source of variation in M. tuberculosis (Mtb) [14]. Through whole-genome sequencing (WGS) analysis of serial samples taken from affected patients, it was recently observed that a significant number of the genetic variations in Mtb are greatly influenced by the host environment [15,16]. In this sense, genes related to DNA damage repair (GRDDR) in Mtb have been the subject of increasing interest in recent years [17], since they play a fundamental role in maintaining DNA integrity and genomic diversification of Mtb [18,19], mainly in damage generated by the environment in which the bacterium is found, and by the host immune response. Mtb presents multiple DNA damage repair mechanisms, some redundant and others specifically related to induced mutagenesis [17]. It has been documented that the presence of SNPs in some Mycobacterium GRDDR can increase the mutation rate [20] and improve the adaptability of the bacteria [21] and can also be associated with increased DR or drug susceptibility [22].

It remains unknown if T2DM in the host could influence the generation of SNPs in Mtb GRDDRs and consequently drug resistance. Therefore, the aim of this study is to compare the presence of SNPs in GRDDRs in sensitive and drug-resistant Mtb genomes isolated from patients with and without T2DM.

## 2. Materials and Methods

### 2.1. Genome Compilation

A search for Mtb genomes was performed in the following public repositories: GenBank, TB Portals, ENA, and Patric. Additionally, a collection of genomes provided by Dr. Iñaki Comas of the Tuberculosis Genomics Unit of the Instituto de Biomedicina de Valencia, Spain, was also considered.

Inclusion of the genomes in the analysis was contingent on the metadata information meeting the following criteria: (1) presence or absence of T2DM in the host, (2) detailed description of the genotypic profile of resistance to first-line drugs (rifampicin, isoniazid, pyrazinamide and ethambutol), (3) coverage values > 99% and depth > 100×, and (4) exclusively belonging to L4, due to its high frequency and global distribution [23]. The final sample consisted of 399 genomes, which were organized into sensitive and drug-resistant TB from hosts with and without T2DM (Appendix A).

### 2.2. Bioinformatics Analysis of Genomes

First, the low-quality ends in the sequences (<30) were trimmed using Fastp [24], then Kraken V.2 [25,26] was used to filter reads belonging to the MTBC complex and avoid false variants as a result of DNA contamination. The reads were aligned with the BWA program [27] using a MTBC reference sequence [28], considering the default parameters. Variant calling (SNPs and INDELS) was performed following a previously described and validated pipeline [5,29], which is available online at http://tgu.ibv.csic.es/?page_id=1794, accessed on 4 October 2021. Variants present in at least 20 reads and at ≥90% frequency within each isolate were used to detect phylogenetic mutations and confirm pertinence to L4. In contrast, variants in at least 10 reads with a frequency of ≥10% to ≤90% were termed non-fixed SNPs and used to detect the first- and second-line drug resistance profile.

From the call of variants previously obtained for each one of the genomes, variants with an allelic frequency ≥ 10% in the coding regions of the 65 genes related to DNA damage repair were identified and selected (Appendix A). A database was then constructed with the non-synonymous SNPs identified in the 65 GRDDRs, the resistance profiles, and the presence/absence of DMT2 in the host.

### 2.3. Statistical Testing, Clustering, and Discriminant Analysis

To identify differences between resistant and sensitive TB genomes from hosts with and without T2DM, and the various sub-lineages comprising the sample, clustering analyses were performed on the non-synonymous SNPs in the GRDDRs only using the IQ-TREE software [30] (http://iqtree.cibiv.univie.ac.at/, accessed on 1 November 2021) with the default parameters for binary data. The generated consensus trees were visualized using iTOL [31] (https://itol.embl.de/, accessed on 1 November 2021).

SNPs present in >99% of genomes were considered as redundant sites. Identification of sub-lineage-related SNPs was performed using the fixation index (Fts = 1), which indicates that the SNP is fixed to the sub-lineage and is not present outside of it. The fixation index was calculated using the Genepop package for Rstudio [32], excluding mixed infections and single sub-lineages (*n* = 1).

Inter-group differences were analyzed by chi-square test using IBM SPSS V21 [33] (95% confidence level), excluding redundant SNPs and those fixed in sub-lineages.

For the discriminant analysis, two functions were generated; one to discriminate the presence or absence of T2DM in the host and another to determine DR-TB and sensibility. Both models were developed using the presence or absence of non-synonymous SNPs in each of the 65 GRDDRs analyzed (0 for absence and 1 for presence of SNPs in the gene). For this analysis, redundant SNPs and those fixed in sub-lineages were excluded, as well as genomes that did not present SNPs other than those related to sub-lineages. The eigenvalue, canonical correlation and Wilks’ lambda were calculated as summary values for each discriminant function. Finally, the generated functions were validated by classifying one case (cross-validation) and a 10% random sample (random sample validation) after exclusion from the discriminant function calculation. SPSS^®^ V21 [33] software was used to perform this analysis and the validation.

## 3. Results

### 3.1. Population Characteristics

A total of 399 Mtb genomes were recovered, with collection periods ranging from 2010 to 2019, from 10 countries, predominantly Georgia (33.6%), Moldova (13.8%), and Indonesia (12.8%). The individuals carrying these isolates were a mean age of 42.5 years (±15.1), 226 (61.4%) were male, and 175 (43.8%) presented T2DM. A total of 223 strains (55.9%) were classified as sensitive, and 176 (44.1%) were classified as mono-resistant, poly-resistant, multi-drug resistant (MDR-TB), pre-extreme-drug resistant (pre-XDR-TB), or extreme-drug resistant (XDR-TB). Resistance to isoniazid was observed in 35%, to rifampicin in 32%, and to ethambutol in 21%. Phylogenetic analysis confirmed that all strains belonged to L4. Nineteen sub-lineages were identified, of which the most frequent were 4.1 (25.3%), 4.1.2.1 (20.3%), and 4.3.3 (17.8%). Mixed infection by L4 strains was found in 2.3% of the genomes (Table 1).

Regarding the analysis groups, 224 genomes belonged to hosts without T2DM, of which 123 (54.9%) had drug sensitive TB and 101 (45.1%) had DR-TB. On the other hand, 175 genomes were from individuals with T2DM, of which 100 (57.1%) had drug sensitive TB and 75 (42.9%) had DR-TB. 

In the 65 GRDDRs analyzed, 352 non-synonymous SNPs were identified. Of these, 346 (98.3%) generated amino acid change, 5 (1.4%) generated early stop codons, and 1 (0.3%) caused stop codon loss. In addition, 241 SNPs (68.5%) were identified as unique in the genomes and 28 SNPs (7.9%) had a frequency ≥ 5% of the sample.

### 3.2. SNPs in Genes Related to DNA Damage Repair

The genes with the highest number of non-synonymous SNPs were DnaE2 (18 SNPs), RecC (14 SNPs), and LigD (14 SNPs), whereas the genes Dut, Ku, Prim-PolC, RecR, and SSBa presented only one non-synonymous SNP. When contrasting the length of the genes, a positive correlation was observed in which the greater the length of the gene, the greater the number of non-synonymous SNPs (r = 0.639, *p* < 0.01). This is evidence that nonsynonymous SNPs are variable among GRDDRs, and this seems to be related to their length.

Ninety-nine percent of the genomes analyzed presented SNPs in the genes AlkA (position 1479085G>A), AdnA (3577958C>T), and ImuA (3811629T>C) (Appendix A). Additionally, <1% of the genomes presented non-synonymous SNPs in SSBa, SSBb, Ku, RecR, RecX, Mpg, DinB2, MazG, Prim-PolC, PolD2, RuvA, RuvC, MutT1, MutT4, and Ung. Notably, RuvX, which encodes a protein responsible for repairing branched DNA structures (Holliday structures) [17], was the only gene lacking non-synonymous SNPs. Likely, the SNPs observed with a frequency > 99% of the sample were previously acquired by a shared ancestor among the different L4 members. On the other hand, genes with lower presence of non-synonymous SNPs could suggest better function or some advantage in terms of conservation.

### 3.3. Cluster Analysis

The dendrogram created with the 352 non-synonymous SNPs identified in the GRDDRs showed that the clusters generated were in complete agreement with the 19 sub-lineages that comprised the sample, but no association was observed with the presence of T2DM in the host (Figure 1). Only lineages 4.2.1 and 4.10 showed differences in the presence of some clusters observed between sensitive and drug resistant genomes, reflected in the presence of non-synonymous SNPs in RecGwed (3011653A>G) and MutT2 (1286927C>A), respectively. The only genome belonging to lineage 4.6.1.1 was placed within the 4.3 lineage group due to the similarity between these lineages. Through the selection of 16 non-synonymous SNPs in 13 GRDDRs (Fts = 1) (Table 2), it was possible to generate a dendrogram that more clearly defined the sub-lineages comprised by the sample (Figure 2). This indicates that certain SNPs in GRDDR are specific to the sub-lineages and strongly related to their evolutionary development.

Given this observation, both sub-lineage-fixed SNPs and redundant SNPs in the sample were given no further consideration in the comparative analysis between host groups or resistance profiles. It should be noted that 11.7% of the genomes (*n* = 47) did not present non-synonymous SNPs, other than those related to the sub-lineage.

### 3.4. Comparison between Drug Sensitive and Drug Resistant Genomes of T2DM and Non-T2DM Hosts

With respect to drug resistance, the distribution of SNPs showed significant differences among groups (Table 3). In the genomes of drug resistant isolates, no non-synonymous SNPs were present in the genes RecR, DinB2, Prim-PolC, RNaseH1, RNaseH2, ImuB, or MutT2, but a high frequency of non-synonymous SNPs was found in RecGwed (*p* < 0.001) (only present in L4.2.1 and mixed infection), MutY (*p* < 0.001), and UvrA (*p* = 0.003). In contrast, drug sensitive TB isolates showed an absence of non-synonymous SNPs in the genes Mpg, SSBa, and Ku, but a high frequency was observed in ImuB (*p* = 0.028), RNaseH2 (*p* = 0.028), RNaseH1 (*p* = 0.046), and MutT2 (*p* < 0.001) (only present in L4.10 and mixed infection).

In relation to the presence/absence of DMT in the host, significant differences were observed in the distribution of genes with SNPs (Table 3). The genomes of individuals with T2DM presented no non-synonymous SNPs in the genes Fpg2 or Ung, but showed a higher frequency of non-synonymous SNPs in LigB (*p* = 0.006), ImuA (*p* = 0.024) (only present in L4.3 and L4.3.3), Prim-PolC (*p* = 0.049) (only present in L4.4.1.1), MazG (*p* = 0.049), RecX (*p* = 0.049) (only present in L4.10), LigD (*p* = 0.006), RuvB (*p* = 0.008), and RecG (*p* = 0.015). The genomes of individuals without T2DM had no non-synonymous SNPs in the genes Prim-PolC, MazG, RecX, RecR, SSBa, and Ku, but showed a higher frequency of non-synonymous SNPs in Cho (*p* < 0.001), PolA (*p* < 0.001), Nei2 (*p* < 0.001), RecGwed (*p* = 0.013), MutT2 (*p* = 0.008), and RecO (*p* = 0.001). This indicates that the occurrence of SNPs in GRDDR is influenced by the presence or absence of T2DM in the host, with a higher presence in isolates from individuals with T2DM. 

### 3.5. Discriminant Analysis

Discriminant function analysis was performed on the 333 non-synonymous SNPs present in the 65 GRDDR genes, assigning values of 0 and 1, respectively, to the absence and presence of non-synonymous SNPs in the gene. Sub-lineage-related SNPs and redundant SNPs in the sample were excluded from the analysis, as well as those genomes that had no SNPs other than those related to sub-lineages. The discriminant function for TB from individuals with or without T2DM showed a moderate but statistically significant discrimination value (eigenvalue = 0.499, canonical correlation = 0.577, Wilks’ lambda = 0.667, df = 63, *p* = 0.000), with functions at the centroids to differentiate between genomes from TB isolates from hosts without T2DM = 0.600 and genomes from TB isolates from hosts with T2DM = −0.827. This analysis correctly classified 73.6% of the genomes of TB from hosts with T2DM, with a sensitivity of 0.69 and a specificity of 0.80. However, it presented lower accuracy in the cross-validation (64.2% correct classification) and validation through a random sample including 10% of the genomes (64.4% correct classification) (Table 4A). From a selection of 63 Mtb GRDDRs, we can duplicate the model (Table 5A), establishing the presence of SNPs in RecR, LigB, and DnaE1 as the most discriminant genes associated with the presence of T2DM in the host.

On the other hand, the discriminant function for sensitive and drug resistant isolates showed moderate but significant values (eigenvalue = 0.788, Canonical correlation = 0.664, Wilks’ lambda = 0.559, df = 63, *p* = 0.000), with functions at the centroids to differentiate between genomes from drug sensitive TB = −0.803 and genomes with drug resistant TB = 0.975. This discriminant function correctly classified 74.8% of the resistant genomes with sensitivity and specificity values of 0.82 and 0.81, respectively (Table 4B), while the cross-validation presented a classification that was 64.8% correct and the random sample validation was 68.1% correct. The model generated with a selection of 63 GRDDRs in the drug resistant and drug sensitive isolates shows that the presence of non-synonymous SNPs in the gene RecGwed had the highest discriminant value, followed by MutY, DnaE2, and Mfd (Table 5B). This demonstrates their high frequency in genomes with some level of drug resistance and thus their utility as discriminants.

Finally, it is important to note that the differences observed between the proportions of isolates correctly classified by the discriminant functions and their validations confirms that the sample used was limited, but still generates functional models with which it is possible to determine the presence of pharmacological resistance and T2DM in the host through analysis of these genes.

## 4. Discussion

Analysis of the polymorphisms in GRDDR in a set of Mtb genomes (lineage 4) allowed the identification of several relationships among the identified SNPs, presence of T2DM in the host, pharmacological resistance and, unexpectedly, the L4 sub-lineages. The results represent the first approach to the study of GRDDRs and evidence an unexpected diversification of SNPs in these Mtb genes, as well as the possible influence of T2DM and DR on their development.

Some of the SNPs identified were highly specific for L4 sub-lineages and could be of further use for classification, while some Mtb genotyping systems refer to the use of certain SNPs from GRDDR, such as: UvrD2 (3570528C>G) [39], MutT3 (500224G>T), MutT4 (4393839C>T), MutY (4031203C>A) [40], DnaE1 (1750465T>C), RuvB (2923264G>A), DinB2 (3416734G>A), and SSBa (58786G>C) [41]. This is the first report of a new panel of 16 non-synonymous SNPs distributed in 13 GRDDRs that allows a correct sub-clustering of L4 sub-linages (Table 2). Although the limited size of the analyzed sample and the low representativeness of some sub-lineages are recognized, the clustering capacity observed with the SNPs present in GRDDR poses new questions about the coevolution of these mutations and the sub-lineages with which they are associated, and raises the possibility of their use as markers for phylogenetic classification.

It has been documented that the genes MutY and UvrA play an important role in DNA repair pathways and that their deficiency sensitizes Mtb to different clastogens [42,43,44]. While MutM (Fpg) and MutY are involved in the cleavage of oxidized guanine and its paired base [20], UvrA (with UvrB) plays an essential role in DNA damage recognition and initiates nucleotide excision repair [17]. On the other hand, RecGwed has a binding affinity for branched DNA structures, mainly in the stationary phase [37], and is involved in drug resistance acquisition pathways of the L4.2.1 [45]. Considering the above, the presence of a more significant number of SNPs in these genes in drug resistant strains could indicate that they result of positive selection induced by anti-TB treatment, which confers advantages when facing the effects of the drugs. On the other hand, the higher frequency of SNPs in the MutT2 (only present in L4.10 and mixed infection), RNaseH1, RNaseH2, and ImuB genes predominantly found in sensitive strains raises several questions regarding their origin and utility.

Differences in the non-synonymous SNPs in the GRDDR observed between individuals with and without T2DM suggest that the presence of T2DM could influence a greater diversity of polymorphisms in the GRDDR in Mtb. This could be explained by the influence of the metabolic and immune system alterations characteristic of patients with T2DM [16,46], reduced plasma concentrations of anti-tuberculosis drugs [8,47], consumption of drugs for glycemic control [9], and the increased formation of cavitary lesions [46], factors that also promote drug resistance [8,46], some of them due to the appearance of mutations generated by the increase of oxidative stress in bacteria [9].

In this sense, the higher frequency of SNPs in LigB ligase and multifunctional LigD ligase, which acts with Ku in the repair of double-strand breaks by non-homologous end joining [48]; Prim-PolC polymerase, which participates in abasic site filling [49]; the accessory protein ImuA, involved in DnaE2-induced mutagenesis [50]; MazG, the protein product of which degrades 5-OH-dCTP to prevent C>T (G>A) mutation when incorporated into DNA [51]; RecX, which regulates the functions of the RecA gene [17], which is fundamental to the SOS response, induced mutagenesis and drug resistance [52] and the RecG and RuvB genes, involved in the resolution of branched DNA structures [53], could explain the increased risk of individuals with the TB-T2DM binomial generating drug resistance or other negative outcomes to anti-tuberculosis treatment. However, further studies are required to determine the specific participation of these or other genes in the process of drug resistance in individuals with the TB-T2DM binomial.

The variations observed between the SNPs of sensitive and resistant isolates from individuals with or without T2DM made it possible to develop two discriminant functions with which to distinguish these characteristics. Besides providing the first description of their use in this context, these functions showed acceptable levels of sensitivity and specificity, which were higher than the random classification values (0.5) in terms of identifying both drug resistant isolates and those from hosts with T2DM. Although there are molecular markers for the diagnosis of drug resistance with better levels of sensitivity and specificity [54], we consider that the discriminant function for drug resistant strains using GRDDR non-synonymous SNPs could be used as a surrogate marker, since it does not use genes commonly related to drug resistance.

We consider that the misclassifications generated by both functions could be influenced by factors related to the time of evolution of the infection, different levels of drug resistance, and clinical control of T2DM in the host. Nevertheless, the presence of non-synonymous SNPs in GRDDR could be used as a tool to distinguish the presence or absence of T2DM in the host, based on Mtb genome analysis. However, further studies are required to confirm the true utility of this proposal.

The main limitation of this study was related to the reduced number of genomes and, consequently, of the members forming the study groups, which acted to generate an unpaired distribution in the sample. This number was related to the limited and non-uniform clinical and epidemiological information of the hosts in the databases. In most cases, information regarding T2DM is limited to presence/absence without incorporating pharmacological treatment for glycemic control, glycemic testing, time elapsed with T2DM condition, and presence of other comorbidities/addictions, which also limited the depth of analysis. Based on the above, incorporation of sufficient clinical and epidemiological information pertaining to the host should be considered a mandatory requirement for the registration of genomes in the repositories. Availability of this information will undoubtedly be key for subsequent studies.

## 5. Conclusions

The present study is the first analysis of polymorphisms present in GRDDR in the context of drug resistance and T2DM. The results show differences in the distribution of non-synonymous SNPs present in GRDDRs that were related to the resistance profile of the infecting strain and the presence of T2DM in the host. This provides evidence that the environment of a patient with T2DM influences the development of SNPs that could be involved in the development of drug resistance. However, further studies are required in order to confirm this possibility.

## Figures and Tables

**Figure 1 genes-13-00609-f001:**
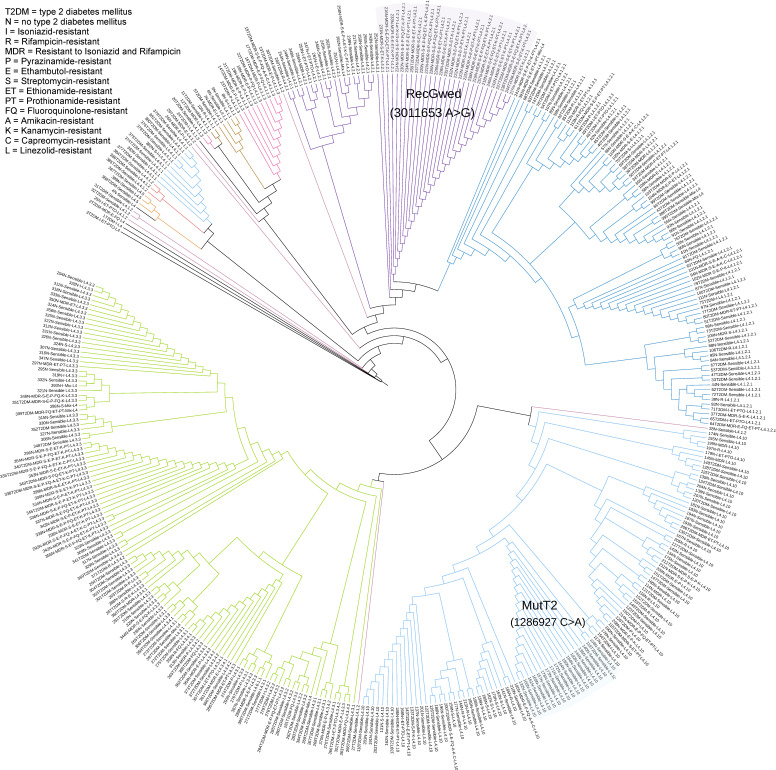
Dendrogram generated using 534 non-synonymous SNPs in genes related to DNA damage repair.

**Figure 2 genes-13-00609-f002:**
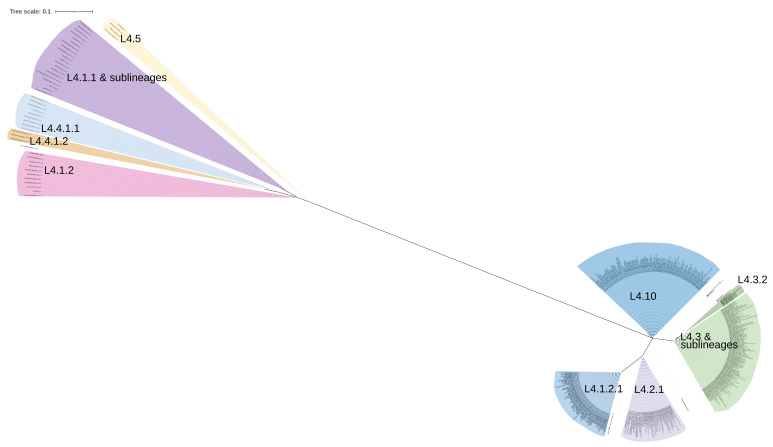
Sub-lineage clustering using 16 non-synonymous SNPs in 13 genes related to DNA damage repair.

**Table 1 genes-13-00609-t001:** Sociodemographic and epidemiological characteristics of the individuals comprising the group of genomes.

	Host without T2DM		Total without T2DM (*n* = 224) *n* (%)	Host with T2DM		Total with T2DM (*n* = 175) *n* (%)	Total Sensitive Isolate (*n* = 223) *n* (%)	Total Drug Resistant Isolate (*n* = 176) *n* (%)	Total (*n* = 399) *n* (%)
	Sensitive Isolate (*n* = 123) *n* (%)	Drug Resistant Isolate (*n* = 101) *n* (%)		Sensitive Isolate (*n* = 100) *n* (%)	Drug Resistant Isolate (*n* = 75) *n* (%)				
Sex *,†									
Female	40 (32.5)	34 (33.7)	74 (33.0)	38 (50.0)	30 (44.1)	68 (47.2)	78 (39.2)	64 (37.9)	142 (38.6)
Male	83 (67.5)	67 (66.3)	150 (67.0)	38 (50.0)	38 (55.9)	76 (52.8)	121 (60.8)	105 (62.1)	226 (61.4)
Age *,†									
Mean (±SD)	38.5 (±15.6)	36.8 (±14.3)	37.7 (±15.0)	50.2 (±11.8)	50.4 (±11.8)	50.3 (±11.8)	42.8 (±15.3)	42.2 (±14.9)	42.5 (±15.1)
Country †									
Georgia	90 (73.2)	37 (36.6)	127 (56.7)	0 (0.0)	7 (9.3)	7 (4.0)	90 (40.4)	44 (25.0)	134 (33.6)
Moldova	19 (15.4)	27 (26.7)	46 (20.5)	3 (3.0)	6 (8.0)	9 (5.1)	22 (9.9)	33 (18.8)	55 (13.8)
Indonesia	0 (0.0)	0 (0.0)	0 (0.0)	42 (42.0)	9 (12.0)	51 (29.1)	42 (18.8)	9 (5.1)	51 (12.8)
Mexico	7 (5.7)	11 (10.9)	18 (8.0)	6 (6.0)	23 (30.7)	29 (16.6)	13 (5.8)	34 (19.3)	47 (11.8)
Spain	0 (0.0)	0 (0.0)	0 (0.0)	27 (27.0)	8 (10.7)	35 (20.0)	27 (12.1)	8 (4.5)	35 (8.8)
Others ^a^	7 (5.7)	26 (25.7)	33 (14.7)	22 (22.0)	22 (29.3)	44 (25.1)	29 (13.0)	48 (27.3)	77 (19.3)
Lineage †									
4.1	48 (39.0)	24 (23.8)	72 (32.1)	19 (19.0)	10 (13.3)	29 (16.6)	67 (30.0)	34 (19.3)	101 (25.3)
4.1.2.1	24 (19.5)	12 (11.9)	36 (16.1)	27 (27.0)	18 (24.0)	45 (25.7)	51 (22.9)	30 (17.0)	81 (20.3)
4.3.3	31 (25.2)	22 (21.8)	53 (23.7)	8 (8.0)	10 (13.3)	18 (10.3)	39 (17.5)	32 (18.2)	71 (17.8)
4.2.1	11 (8.9)	29 (28.7)	40 (17.9)	0 (0.0)	7 (9.3)	7 (4.0)	11 (4.9)	36 (20.5)	47 (11.8)
4.1.1.3	0 (0.0)	2 (2.0)	2 (0.9)	1 (1.0)	10 (13.3)	11 (6.3)	1 (0.4)	12 (6.8)	13 (3.3)
L4 Mix-inf	1 (0.8)	3 (3.0)	4 (1.8)	4 (4.0)	1 (1.3)	5 (2.9)	5 (2.2)	4 (2.3)	9 (2.3)
Others ^b^	8 (6.5)	9 (8.9)	17 (7.6)	41 (41.0)	19 (25.3)	60 (34.3)	49 (22.0)	28 (15.9)	77 (19.3)
Resistance profile									
Sensitive	123 (100.0)	-	123 (54.9)	100 (100.0)	-	100 (57.1)	223 (100.0)	-	223 (55.9)
Mono-resistant	-	21 (20.8)	21 (9.4)	-	20 (26.7)	20 (11.4)	-	41 (23.3)	41 (10.3)
Poly-resistant	-	11 (10.9)	11 (4.9)	-	13 (17.3)	13 (7.4)	-	24 (13.6)	24 (6.0)
MDR	-	20 (19.8)	20 (8.9)	-	18 (24.0)	18 (10.3)	-	38 (21.6)	38 (9.5)
Pre-XDR	-	31 (30.7)	31 (13.8)	-	17 (22.7)	17 (9.7)	-	48 (27.3)	48 (12.0)
XDR	-	18 (17.8)	18 (8.0)	-	7 (9.3)	7 (4.0)	-	25 (14.2)	25 (6.3)

T2DM: Type 2 diabetes mellitus. MDR: Multidrug Resistant. XDR: Extensively Drug Resistant. Mix-inf: Mixed infection. * Patients without information are excluded. † Statistically significant difference between hosts (chi square test, *p* < 0.05). ^a^ Other countries: Peru, Romania, Belarus, Azerbaijan, and Kazakhstan. ^b^ Other lineages: lineage 4.3.1, 4.3.4.2, 4.4.1.1, 4.3.2, 4.1.2, 4.3.4.1, 4.1.1, 4, 4.4.1.2, 4.5, 4.1.1.1, 4.3, 4.2.2, and 4.6.1.1.

**Table 2 genes-13-00609-t002:** Nonsynonymous SNPs in DNA damage repair-related genes used to classify sub-lineages.

Gene	Gene Function	Site	Nucleotide Change	Sub-Lineage
*RecC*	Participates in the single-strand annealing pathway [17]	726703	G>T	4.3, 4.3.3, 4.3.1, 4.3.4.1, and 4.3.4.2
		726816	C>G	4.1.1, 4.1.1.1, and 4.1.1.3
*LigD*	DNA ligase [17]	1047165	C>T	4.10
		1047683	G>T	4.3.2
*Mfd*	Recognizes transcription problems and recruits UvrABC [17]	1139102	G>A	4.1.1.1
*Ogt/adaB*	Repairs alkylated guanine in DNA [34]	1477588	C>G	4.1.2.1
*DinB1*	DNA polymerase [17]	1740771	A>C	4.3, 4.3.3, 4.3.1, 4.3.4.1, and 4.3.4.2
*UvrB*	Participates in recognition of DNA damage (with UvrA) and initiates nucleotide excision repair [35]	1838153	G>A	4.3.2
*Nei1*	Excises oxidized pyrimidines [36]	2767631	G>A	4.4.1.1
*RecGwed*	Binds to branched DNA structures [37]	3011692	T>G	4.1.2.1
*Dut*	Involved in Nucleotide Pool Sanitization [17]	3013784	G>C	4.1.2.1
*LigB*	DNA ligase [17]	3426025	G>A	4.5
*AdnB*	Involved in repair by homologous recombination [17]	3574504	C>T	4.2.1 and 4.2.2
		3575106	T>C	4.4.1.1
*DnaE2*	Error-prone DNA polymerase [38]	3781574	C>G	4.4.1.2
*LigC*	DNA ligase [17]	4182695	A>G	4.10
*AlkA*	Exhibits methyltransferase activity [17]	1479085	G>A	Present in >99% of the sample
*AdnA*	With AdnB, initiates DNA double-strand break repair by RecA-dependent homologous recombination [17]	3577958	C>T	Present in >99% of the sample
*ImuA*	Encodes a DnaE2 accessory protein [17]	3811629	T>C	Present in >99% of the sample

**Table 3 genes-13-00609-t003:** Distribution of genes with SNPs according to drug resistance and absence/presence of T2DM in the host.

Gene	Host without T2DM		Total without T2DM (*n* = 224) *n* (%)	Host with T2DM		Total with T2DM (*n* = 175) *n* (%)	Total Sensitive Isolate (*n* = 223) *n* (%)	Total Drug-Resistant Isolate (*n* = 176) *n* (%)	Total (*n* = 399) *n* (%)	Exclusive Sub-Linage Diversification
	Sensitive Isolate (*n* = 123) *n* (%)	Drug Resistant Isolate (*n* = 101) *n* (%)		Sensitive Isolate (*n* = 100) *n* (%)	Drug Resistant Isolate (*n* = 75) *n* (%)					
*LigD •*	5 (4.1)	2 (2.0)	7 (3.1)	12 (12.0)	5 (6.7)	* 17 (9.7)	17 (7.6)	7 (4.0)	24 (6.0)	No
*Mfd •*	2 (1.6)	4 (4.0)	6 (2.7)	4 (4.0)	4 (5.3)	8 (4.6)	6 (2.7)	8 (4.5)	14 (3.5)	No
*MazG*	0 (0.0)	0 (0.0)	0 (0.0)	2 (2.0)	1 (1.3)	* 3 (1.7)	2 (0.9)	1 (0.6)	3 (0.8)	No
*MutT2*	17 (13.8)	0 (0.0)	* 17 (7.6)	3 (3.0)	0 (0.0)	3 (1.7)	** 20 (9.0)	0 (0.0)	20 (5.0)	4.10 and Mix-inf
*DnaE1*	4 (3.3)	1 (1.0)	5 (2.2)	4 (4.0)	5 (6.7)	9 (5.1)	8 (3.6)	6 (3.4)	14 (3.5)	No
*PolA*	32 (26.0)	29 (28.7)	* 61 (27.2)	5 (5.0)	8 (10.7)	13 (7.4)	37 (16.6)	37 (21)	74 (18.5)	No
*UvrA*	4 (3.3)	15 (14.9)	19 (8.5)	6 (6.0)	7 (9.3)	13 (7.4)	10 (4.5)	** 22 (12.5)	32 (8.0)	No
*Cho*	41 (33.3)	22 (21.8)	* 63 (28.1)	5 (5.0)	9 (12.0)	14 (8.0)	46 (20.6)	31 (17.6)	77 (19.3)	No
*RNaseH1*	4 (3.3)	0 (0.0)	4 (1.8)	1 (1.0)	0 (0.0)	1 (0.6)	** 5 (2.2)	0 (0.0)	5 (1.3)	No
*RecO*	7 (5.7)	9 (8.9)	* 16 (7.1)	1 (1.0)	0 (0.0)	1 (0.6)	8 (3.6)	9 (5.1)	17 (4.3)	No
*RuvB*	1 (0.8)	2 (2.0)	3 (1.3)	8 (8.0)	3 (4.0)	* 11 (6.3)	9 (4.0)	5 (2.8)	14 (3.5)	No
*RecGwed •*	0 (0.0)	24 (23.8)	* 24 (10.7)	1 (1.0)	6 (8.0)	7 (4.0)	1 (0.4)	** 30 (17)	31 (7.8)	4.2.1 and Mix-inf
*RecX*	0 (0.0)	0 (0.0)	0 (0.0)	2 (2.0)	1 (1.3)	* 3 (1.7)	2 (0.9)	1 (0.6)	3 (0.8)	4.10.
*RNaseH2*	4 (3.3)	0 (0.0)	4 (1.8)	2 (2.0)	0 (0.0)	2 (1.1)	** 6 (2.7)	0 (0.0)	6 (1.5)	No
*RecG*	2 (1.6)	1 (1.0)	3 (1.3)	8 (8.0)	2 (2.7)	* 10 (5.7)	10 (4.5)	3 (1.7)	13 (3.3)	No
*LigB •*	1 (0.8)	0 (0.0)	1 (0.4)	6 (6.0)	2 (2.7)	* 8 (4.6)	7 (3.1)	2 (1.1)	9 (2.3)	No
*Nei2*	28 (22.8)	22 (21.8)	* 50 (22.3)	4 (4.0)	8 (10.7)	12 (6.9)	32 (14.3)	30 (17)	62 (15.5)	No
*DnaE2 •*	15 (12.2)	21 (20.8)	36 (16.1)	20 (20.0)	17 (22.7)	37 (21.1)	35 (15.7)	38 (21.6)	73 (18.3)	No
*ImuB*	2 (1.6)	0 (0.0)	2 (0.9)	4 (4.0)	0 (0.0)	4 (2.3)	** 6 (2.7)	0 (0.0)	6 (1.5)	No
*ImuA •*	1 (0.8)	0 (0.0)	1 (0.4)	4 (4.0)	2 (2.7)	* 6 (3.4)	5 (2.2)	2 (1.1)	7 (1.8)	4.3 and 4.3.3 (site 3811327)
*MutY*	0 (0.0)	17 (16.8)	17 (7.6)	2 (2.0)	6 (8.0)	8 (4.6)	2 (0.9)	** 23 (13.1)	25 (6.3)	No
*RecR*	0 (0.0)	0 (0.0)	0 (0.0)	2 (2.0)	0 (0.0)	2 (1.1)	2 (0.9)	0 (0.0)	2 (0.5)	No
*Prim-PolC*	0 (0.0)	0 (0.0)	0 (0.0)	3 (3.0)	0 (0.0)	* 3 (1.7)	3 (1.3)	0 (0.0)	3 (0.8)	4.4.1.1

T2DM: Type 2 diabetes mellitus. Only genes with significant differences between groups are presented. • Lineage-related SNPs and SNPs > 99% of the sample are excluded. * Statistically significant difference between hosts with/without T2DM (chi-square test, *p* < 0.05). ** Statistically significant difference between sensitive and drug-resistant strains (chi-square test, *p* < 0.05). Genes not shown in the table are described in Appendix A.

**Table 4 genes-13-00609-t004:** Discriminant analysis to predict diversification associated with the presence of T2DM in the host and drug resistance according to SNPs in genes related to DNA damage repair.

	Predicted Group		Total	Classified Correctly	Sensitivity	Specificity	Positive Predictive Value	Negative Predictive Value
Group	Host with T2DM *n* (%)	Host without T2DM *n* (%)						
(A) Original								
Host with T2DM	109 (73.6)	39 (26.4)	148	75.3%	0.69	0.80	74%	76%
Host without T2DM	48 (26.5)	156 (69.6)	204					
Cross-validation *								
Host with T2DM	94 (63.5)	54 (36.5)	148	64.2%	0.57	0.71	64%	65%
Host without T2DM	72 (35.3)	132 (64.7)	204					
Random sample validation (10%) **	-	-	-	64.4%	-	-	-	-
(B) Original	Drug-resistant *n* (%)	Sensitive *n* (%)	Total					
Drug-resistant	119 (74.8)	40 (25.2)	159	81.3%	0.82	0.81	75%	87%
Sensitive	26 (13.5)	167 (86.5)	193					
Cross-validation *								
Drug-resistant	98 (61.6)	61 (38.4)	159	64.8%	0.61	0.68	62%	67%
Sensitive	63 (32.6)	130 (67.4)	193					
Random sample validation (10%) **	-	-	-	68.1%	-	-	-	-

T2DM: Type 2 diabetes mellitus. * In cross-validation, each case is classified using the functions derived from the rest of the cases. ** 10% of the sample is excluded from the function calculation and is classified using the functions derived from the rest of the cases; the average obtained from 10 repetitions is presented. The standardized coefficients for both discriminant functions are detailed in Table 5.

**Table 5 genes-13-00609-t005:** Standardized coefficients of the canonical discriminant function for hosts with/without T2DM and drug-resistant TB. (**A**) Negative values indicate the presence of T2DM in the host and positive values indicate the absence of T2DM in the host. (**B**) Positive values indicate drug resistance and negative values indicate drug-susceptible TB.

(A) Discriminant Function Coefficients for Hosts with/without T2DM						(B) Discriminant Function Coefficients for Drug-Resistant and Sensitive TB					
Gene	Coefficient	Gene	Coefficient	Gene	Coefficient	Gene	Coefficient	Gene	Coefficient	Gene	Coefficient
*RecR*	−0.422	*UvrD2*	−0.052	*MutT4*	0.076	*RecGwed*	0.777	*RuvB*	0.114	*RadA*	−0.055
*LigB*	−0.414	*RNaseH2*	−0.051	*DinB1*	0.08	*MutY*	0.609	*End (Nfo)*	0.074	*AdnB*	−0.065
*DnaE1*	−0.243	*RuvA*	−0.043	*AdnA*	0.093	*DnaE2*	0.329	*UvrD2*	0.072	*RecN*	−0.072
*LigD*	−0.237	*MutT1*	−0.043	*Nei1*	0.1	*Mfd*	0.305	*RecA*	0.068	*LigA*	−0.077
*MazG*	−0.233	*PolD2*	−0.038	*TagA*	0.103	*XthA*	0.3	*MutT1*	0.031	*RecX*	−0.089
*ImuA*	−0.203	*ImuB*	−0.034	*Rv2119*	0.103	*RecF*	0.291	*UvrC*	0.031	*RecC*	−0.097
*End (Nfo)*	−0.194	*SSBb*	−0.032	*UdgB*	0.104	*UvrB*	0.249	*PolA*	0.028	*RuvA*	−0.098
*SSBa*	−0.173	*XthA*	−0.03	*RecD*	0.111	*TagA*	0.24	*SSBb*	0.026	*MutT3*	−0.099
*Cho*	−0.164	*Mfd*	−0.009	*Ogt/adaB*	0.12	*Mpg*	0.222	*Ung*	0.021	*Fpg2*	−0.11
*MutT3*	−0.151	*RecC*	0.001	*Nth*	0.129	*NucS*	0.221	*AdnA*	0.017	*RecG*	−0.137
*RecX*	−0.144	*RecN*	0.006	*UvrD1*	0.134	*AlkA*	0.216	*UvrA*	0.013	*Nth*	−0.157
*MutY*	−0.133	*Mpg*	0.009	*Ung*	0.137	*Nei2*	0.203	*LigD*	0.011	*UvrD1*	−0.162
*AlkA*	−0.129	*DnaE2*	0.01	*DinB2*	0.144	*RecD*	0.195	*RuvC*	0.004	*Nei1*	−0.17
*Prim-PolC*	−0.113	*NucS*	0.02	*LigA*	0.178	*RecO*	0.166	*MutT4*	−0.001	*RNaseH2*	−0.173
*UvrB*	−0.111	*AdnB*	0.025	*Fpg2*	0.181	*Ku*	0.157	*DinB1*	−0.001	*MutT2*	−0.175
*Ku*	−0.108	*RNaseH1*	0.026	*RecB*	0.259	*SSBa*	0.154	*DnaE1*	−0.004	*Cho*	−0.181
*MutM (Fpg)*	−0.096	*UvrA*	0.032	*Nei2*	0.298	*LigC*	0.151	*MazG*	−0.01	*Prim−PolC*	−0.186
*RuvB*	−0.094	*RuvC*	0.037	*RecGwed*	0.363	*UdgB*	0.141	*ImuA*	−0.025	*ImuB*	−0.195
*UvrC*	−0.081	*RecF*	0.057	*RecO*	0.391	*MutM (Fpg)*	0.134	*Rv2119*	−0.033	*RNaseH1*	−0.206
*RecA*	−0.071	*LigC*	0.058	*PolA*	0.501	*PolD2*	0.133	*Ogt/adaB*	−0.044	*RecB*	−0.236
*RecG*	−0.053	*RadA*	0.062	*MutT2*	0.549	*RecR*	0.116	*DinB2*	−0.047	*LigB*	−0.315

## Data Availability

Data is contained within the article or Appendix A.

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
