# Peer review of "SNPs in Genes Related to DNA Damage Repair in Mycobacterium Tuberculosis: Their Association with Type 2 Diabetes Mellitus and Drug Resistance"

_genes, 2022, doi:10.3390/genes13040609_

Round 1

Reviewer 1 Report

While this paper addresses an interesting and important topic – the adaptation of Mtb strains in the context of host metabolic differences due to type 2 DM, there are a number of methodological issues that should be addressed as well as a lack of clarity in the presentation. In general, the study is quite difficult to follow and I believe it would benefit from some restructuring in addition to addressing some of the limitations mentioned below:

  1. The study uses whole genome sequence data available in public repositories that include “meta-data” but it does not specify how these data were originally obtained from study participants. This is of particular importance with regard to the diagnosis of type 2 DM which often goes undetected in people who are not specifically tested for it. Global estimates suggest that half or more of all DM is undiagnosed so it will be important to know if individuals who were classified as having DM or more specifically, not having DM, were actually tested. Since hypergylcemia is a common finding in acutely ill TB patients regardless of their DM status, this testing would have had to be done after the completion of TB treatment to actually reflect true DM status.   Furthermore, among the subset of DM patients who self-report DM, many are undergoing treatment that may include drugs like Metformin which might also have an impact on TB genomes, in different ways than DM would be expected to have. Without these data, it is difficult to know how to interpret the diagnosis of T2DM. Ideally, the study should be limited to isolates from patients for whom such data is known. The comment in the limitations that the small number of genomes “was related to the paucity of information  found in the metadata describing the relevant clinical and epidemiological information of the hosts, such as confirmation of the presence or absence of T2DM, pharmacological treatment for glycemia control, glycemia tests, time elapsed with the T2DM condition, presence of other comorbidities, and presence or absence of addictions,” does not make clear what information was available for the samples used.

  1. The division of isolates into four groups (which I assume are mutually exclusive although that is not entirely clear from the way the methods are written) is interesting but it is not entirely clear to me what hypotheses the authors are testing here. The introduction and methods sections would benefit from more clearly stated hypotheses regarding the expected differences in GRDDRs between the DM and non-DM groups and the DR and DS groups. This is a very small data set to be making multiple cross-group comparisons (DM versus non-DM, DR versus DS, DR/DM versus DR/non-DM, etc.)

  1. Given 65 GRDDRs with an unspecified number of base pairs, there should be a correction for multiple comparisons for which the denominator is the base pair count of non-synonymous sites in these genes and for the discriminant analysis, the number of genes. I suspect that given the very small sample size, adjusting for multiple comparisons in this way will render most of these findings non-significant, which may help streamline the analysis to focus on genes for which there was a stronger signal. As it is now, there is a great deal of descriptive data about individual genes – so much so that it is challenging to interpret. Setting a more stringent criteria for this analysis might lead to a more interpretable result.

  1. The complex descriptive nature of the analyses is also manifest in the tables which contain too much data to be useful or even readable. Some of this can be moved to an appendix and the tables should really highlight results that the authors find important and informative regarding their overall hypotheses.

  1. The way in which the analysts have dealt with lineage markers – by excluding genes in which 99% of samples have snps or snps “strongly related to sublineages” – seems somewhat random. There are a number of ways this has been addressed in previous analysis – PhyC, the use of a genetic relatedness matrix – and the authors should consider the use of these methods.

  1. It is not clear to me why the analysis was restricted to Lineage 4 strains. The authors note that this is because this lineage has no inherent tendency to develop resistance although that point is not referenced and is debatable. In any case, strains that do have a tendency to develop resistance are those most likely to have snps in DNA repair genes so it seems like it would have been useful to include other lineages both to increase the sample size and to observe whether there are consistent snps across lineages associated with DM. this larger analysis could have included this as a sub-analysis by lineage if that seemed interesting given the initial results.

  1. I am not sure I understand the value of identifying snps within GRDDRs that could serve to mark lineages. Lineage identification with whole genomes is not challenging and with the low cost of sequencing and increasing interest in using WGS for drug resistance detection, there is not really a role for targeted sequencing of GRDDRs alone.

Author Response

Comment

reply

1 The study uses whole genome sequence data available in public repositories that include “meta-data” but it does not specify how these data were originally obtained from study participants. This is of particular importance with regard to the diagnosis of type 2 DM which often goes undetected in people who are not specifically tested for it. Global estimates suggest that half or more of all DM is undiagnosed so it will be important to know if individuals who were classified as having DM or more specifically, not having DM, were actually tested. Since hypergylcemia is a common finding in acutely ill TB patients regardless of their DM status, this testing would have had to be done after the completion of TB treatment to actually reflect true DM status.   Furthermore, among the subset of DM patients who self-report DM, many are undergoing treatment that may include drugs like Metformin which might also have an impact on TB genomes, in different ways than DM would be expected to have. Without these data, it is difficult to know how to interpret the diagnosis of T2DM. Ideally, the study should be limited to isolates from patients for whom such data is known. The comment in the limitations that the small number of genomes “was related to the paucity of information  found in the metadata describing the relevant clinical and epidemiological information of the hosts, such as confirmation of the presence or absence of T2DM, pharmacological treatment for glycemia control, glycemia tests, time elapsed with the T2DM condition, presence of other comorbidities, and presence or absence of addictions,” does not make clear what information was available for the samples used.

Thank you for the comment. Ensuring the validity of the information, particularly the presence/absence of T2DM in the host was our main commitment. We agree with the current difficulties in adequately detecting T2DM in a TB patient. Some countries, mainly with high prevalence of both diseases, choose to screen for T2DM in every new TB case over 18 years of age, as well as intensive TB search in people living with T2DM. However, adequate diagnosis of T2DM depends on the capacity of the health system, its regulations and technological reach.

The presence/absence of T2DM in the host for those samples outside Mexico and Spain was obtained directly from the databases. TB-Portals mentions that the presence of T2DM is governed by WHO criteria (without further details), however, it assures that the published data "are de-identified, properly curated and validated to become a trusted accessible resource". For GenBank, metadata is filled in by those who contribute genomes. In the case of genomes from Mexico and Spain the diagnosis of T2DM was made and reported by the health systems.

Despite this, the limited availability and lack of uniformity in the metadata on the time living with T2DM and its treatment did not allow a deeper analysis of the presence/absence of T2DM in the hosts.

Therefore, and following up on their observations, the following information was added in the paragraph that addresses the limitations of the study:

(P-12/L 326-31) “This number was related to the limited and non-uniform clinical and epidemiological information of the hosts in the databases. In most cases, information regarding T2DM is limited to presence/absence without incorporating pharmacological treatment for glycemic control, glycemic testing, time elapsed with T2DM condition, and presence of other comorbidities/addictions, which also limited the depth of analysis.”   

2 The division of isolates into four groups (which I assume are mutually exclusive although that is not entirely clear from the way the methods are written) is interesting but it is not entirely clear to me what hypotheses the authors are testing here. The introduction and methods sections would benefit from more clearly stated hypotheses regarding the expected differences in GRDDRs between the DM and non-DM groups and the DR and DS groups. This is a very small data set to be making multiple cross-group comparisons (DM versus non-DM, DR versus DS, DR/DM versus DR/non-DM, etc.)

Thank you for the comment. We corrected the wording of the method and introduction section to clarify the objectives and analysis groups:

Introduction: (P-2 / L 64-67) “It remains unknown if T2DM in the host could influence the generation of SNPs in Mtb GRDDRs and consequently drug resistance. Therefore, we compared the presence of SNPs in GRDDRs in sensitive and drug-resistant Mtb genomes isolated from patients with and without T2DM.”

Method: (P-2 / L 78-80) “The final sample consisted of 399 genomes, which were organized into sensitive and drug-resistant TB from hosts with and without T2DM.”

3 Given 65 GRDDRs with an unspecified number of base pairs, there should be a correction for multiple comparisons for which the denominator is the base pair count of non-synonymous sites in these genes and for the discriminant analysis, the number of genes. I suspect that given the very small sample size, adjusting for multiple comparisons in this way will render most of these findings non-significant, which may help streamline the analysis to focus on genes for which there was a stronger signal. As it is now, there is a great deal of descriptive data about individual genes – so much so that it is challenging to interpret. Setting a more stringent criteria for this analysis might lead to a more interpretable result.

We appreciate your comments. We agree that the intention of the article is descriptive and that the sample size does not allow for multiple comparisons.

However, considering their observations, several values were eliminated from Table 3 and the descriptive paragraphs related to them.

Additionally, the suggested analyses were performed, and no significant differences were observed.

4 The complex descriptive nature of the analyses is also manifest in the tables which contain too much data to be useful or even readable. Some of this can be moved to an appendix and the tables should really highlight results that the authors find important and informative regarding their overall hypotheses.

Thank you for your comment. We agree that the scope of our publication is descriptive. It is intended to generate a first approximation of the SNPs in GRDDRs in clinical isolates, and the differences observed between hosts with/without T2DM and DR or DS.

At your suggestion, Table 3 and the results section were reduced to focus on the most significant genes. The rest of the genes analyzed are presented in Supplementary table 4. Distribution of genes with SNPs according to drug resistance and absence/presence of T2DM in the host (continuation).

5 The way in which the analysts have dealt with lineage markers – by excluding genes in which 99% of samples have snps or snps “strongly related to sublineages” – seems somewhat random. There are a number of ways this has been addressed in previous analysis – PhyC, the use of a genetic relatedness matrix – and the authors should consider the use of these methods.

Thanks for the comment. We have added in the methodology the selection criteria for redundant SNPs and those fixed in the sublineages.

(P 3 / L 103-107)  ”SNPs present in >99% of genomes were considered as redundant sites. Identification of sublineage-related SNPs was performed using the fixation index (Fts=1), which indicates that the SNP is fixed to the sublineage and is not present outside of it. The fixation index was calculated using the Genepop package for Rstudio[32], excluding mixed infections and single sublineages (n=1).”

Redundant SNPs were eliminated due to their low informative value according to the objective of the study.

6 It is not clear to me why the analysis was restricted to Lineage 4 strains. The authors note that this is because this lineage has no inherent tendency to develop resistance although that point is not referenced and is debatable. In any case, strains that do have a tendency to develop resistance are those most likely to have snps in DNA repair genes so it seems like it would have been useful to include other lineages both to increase the sample size and to observe whether there are consistent snps across lineages associated with DM. this larger analysis could have included this as a sub-analysis by lineage if that seemed interesting given the initial results.

Thanks for the comment. The wording on the selection criterion for lineage 4 was improved:

(p-2 / L-78)"and 4) belonging exclusively to L4, due to its high frequency and global distribution [23]."

We restrict the analysis to Linage 4 because of its wide worldwide distribution, but we agree that it will be interesting to extend this research to other lineages. However, due to external constraints, for the time being we seek to publish only on L4.

7 I am not sure I understand the value of identifying snps within GRDDRs that could serve to mark lineages. Lineage identification with whole genomes is not challenging and with the low cost of sequencing and increasing interest in using WGS for drug resistance detection, there is not really a role for targeted sequencing of GRDDRs alone.

Thank you for your comment. We agree that lineage identification is not a current challenge in epidemiological surveillance systems based on Whole Genome Sequencing. However, given the lack of publications that address the classification of TB sublineages (L4) by means of SNPs present in GRDDRs, we consider it interesting to add it to the results. Some of these results could be used to develop PCR tests to find these variables quickly and perform subrogated lineage assignment.

Reviewer 2 Report

The article is well written and provides interesting results on the influence of type 2 diabetes mellitus (T2DM) on the presence of SNPs in DNA damage repair-related genes and the association with Drug Resistance in M. tuberculosis.
However, I have some minor recommendations:

Some typo errors were noted (e.g. "was also used considered..." in section 2.1, line 73).

Please add more details about the statistical analysis (eigenvalue, Canonical correlation, Wilks' lambda...) and the methodology or software tools used in the Materials and Methods section.

It would be interesting to perform the same analysis regarding SNPs present in GRDDRs for other Mtb lineages (such as L2).
Further details on how to access genomes and their metadata should be provided (GenBank accession numbers, ...).

It would be also interesting to compare results provided in this study for phylogenetic classification of L4 with other existing methods using SNPs.

Author Response

Comment

reply

Some typo errors were noted (e.g. "was also used considered..." in section 2.1, line 73).

Thank you for your comment. The document has been reviewed and all typos found corrected. 

Please add more details about the statistical analysis (eigenvalue, Canonical correlation, Wilks' lambda...) and the methodology or software tools used in the Materials and Methods section.

Thank you for your comment. Details about the statistical analysis and software used was included noy on the methods section.

(p-3 / L 115-117)“The eigenvalue, canonical correlation and Wilks' lambda were calculated as summary values for each discriminant function.”

It would be interesting to perform the same analysis regarding SNPs present in GRDDRs for other Mtb lineages (such as L2).

Thank you for your comment. We restrict the analysis to L4 by considering its worldwide distribution. We totally agree about the need of made this analysis in other lineages such as L1, L2 o L3, at this moment we are addressing this analysis with L2 isolates.  

Further details on how to access genomes and their metadata should be provided (GenBank accession numbers, ...).

Thank you for your comment. We have included all the IDs of the genomes selected and analyzed in this study in the Supplementary table 1. “Data of the analyzed sequences”

It would be also interesting to compare results provided in this study for phylogenetic classification of L4 with other existing methods using SNPs.

We agree with your comment, perhaps the results from the analysis of the other lineages could be included and analyzed to have a better representation of the lineage representation. In addition, as was mentioned, these results were additional to the main goal of the study. Further studies should, considering this recommendation, be addressed promptly.